# Assessment of Ambient Air Toxics and Wood Smoke Pollution among Communities in Sacramento County

**DOI:** 10.3390/ijerph17031080

**Published:** 2020-02-08

**Authors:** Steven G. Brown, Janice Lam Snyder, Michael C. McCarthy, Nathan R. Pavlovic, Stephen D’Andrea, Joseph Hanson, Amy P. Sullivan, Hilary R. Hafner

**Affiliations:** 1Sonoma Technology, Inc., Petaluma, CA 95494, USA; mmccarthy@sonomatech.com (M.C.M.); npavlovic@sonomatech.com (N.R.P.); hilary@sonomatech.com (H.R.H.); 2Sacramento Metropolitan Air Quality Management District (SMAQMD), Sacramento, CA 95814, USA; JLam@airquality.org (J.L.S.); SDAndrea@airquality.org (S.D.); 3Meta Research, Inc., Sacramento, CA 95811, USA; Jhanson2@gmail.com; 4Atmospheric Science Department, Colorado State University, Fort Collins, CO 80521, USA; Amy.Sullivan@ColoState.EDU

**Keywords:** community air monitoring, black carbon, wood smoke, air toxics

## Abstract

Ambient air monitoring and phone survey data were collected in three environmental justice (EJ) and three non-EJ communities in Sacramento County during winter 2016–2017 to understand the differences in air toxics and in wood smoke pollution among communities. Concentrations of six hazardous air pollutants (HAPs) and black carbon (BC) from fossil fuel (BC_ff_) were significantly higher at EJ communities versus non-EJ communities. BC from wood burning (BC_wb_) was significantly higher at non-EJ communities. Correlation analysis indicated that the six HAPs were predominantly from fossil fuel combustion sources, not from wood burning. The HAPs were moderately variable across sites (coefficient of divergence (COD) range of 0.07 for carbon tetrachloride to 0.28 for m- and p-xylenes), while BC_ff_ and BC_wb_ were highly variable (COD values of 0.46 and 0.50). The BC_wb_ was well correlated with levoglucosan (*R*^2^ of 0.68 to 0.95), indicating that BC_wb_ was a robust indicator for wood burning. At the two permanent monitoring sites, wood burning comprised 29–39% of the fine particulate matter (PM_2.5_) on nights when PM_2.5_ concentrations were forecasted to be high. Phone survey data were consistent with study measurements; the only significant difference in the survey results among communities were that non-EJ residents burn with indoor devices more often than EJ residents.

## 1. Introduction

Wood smoke from residential burning is the largest source of wintertime particulate matter with aerodynamic diameter less than 2.5 microns (PM_2.5_) emissions in the Sacramento, California, area, accounting for more than 50 percent of direct wintertime PM_2.5_ emissions [1]. Ambient pollution studies have also indicated that wood smoke is a major source of wintertime PM_2.5_ in Sacramento [2,3]. In Sacramento, this wintertime source of PM_2.5_ is particularly important; residential burning emissions typically occur in the evening or overnight, when they can be trapped in the shallow boundary layer that often forms in the Sacramento Valley. The resultant high-concentration PM_2.5_ events over the course of an evening can have acute health impacts [4,5], and exposure to wood burning emissions have been linked to health effects [5,6,7].

Wood smoke is a mixture of organic carbon and black carbon (BC), secondarily formed organic mass, and a wide range of gaseous species [8,9,10,11]. Wood smoke includes toxics such as formaldehyde, acetaldehyde, and acetonitrile [12,13] as well as acrolein, polycyclic organic matter (POM), benzene, and dioxins [14,15], most of which are listed among the Environmental Protection Agency’s (EPA) 30 Urban Air Toxics of Concern and are leading drivers of risk nationally [16]. Wood smoke from residential biomass burning differs from other pollutant sources in that it can be more localized than urban- or regional-scale pollutants such as ammonium nitrate, sulfate, or organic matter.

While there are thousands of chemicals emitted from residential biomass burning, unique chemicals are emitted from the combustion of wood lignin, namely, levoglucosan and other anhydrous sugars; therefore, levoglucosan was used as the main tracer for wood burning emissions. Levoglucosan is emitted during combustion of wood cellulose [17], and its emissions, relative to total emitted PM, can vary by fuel type and burning condition [18]. Levoglucosan may be oxidized in the atmosphere [19,20], but is relatively stable compared to other co-emitted compounds and is emitted from biomass burning in relative abundance, making it a commonly used wood smoke tracer [21,22]. In addition to direct filter-based measurements of levoglucosan, multi-channel Magee Scientific Aethalometers have been used to determine black carbon from wood burning (BC_wb_) and from fossil fuel combustion (BC_ff_) [23]; this method has been used widely in Europe and at a handful of locations in the United States [24,25,26,27,28,29].

Pollution can be higher in environmental justice (EJ) areas which are typically lower-income areas with a higher proportion of minority residents compared to non-EJ areas [30,31,32,33]. The US EPA defines EJ as the fair treatment and meaningful involvement of all people regardless of race, color, national origin, or income with respect to the development, implementation, and enforcement of environmental laws, regulations, and policies. The EPA has developed a tool, EJ Screen, that produces EJ indices for each census block in the US by combining environmental indicators, such as proximity to freeways, with demographic indices such as income level. In Sacramento, multiple communities are in the highest 5% EJ Index nationally for both PM_2.5_ and cancer risk. This study focused on three such communities—South Natomas, Arden, and South Sacramento—and paired them with nearby non-EJ communities—T Street, Del Paso Manor, and Colonial Heights. Since wood smoke from residential burning is the largest single source of wintertime PM_2.5_ in Sacramento, and is typically due to local rather than urban- or regional-scale sources, we investigated the levels of PM_2.5_, hazardous air pollutants (HAPs), and wood smoke contributions to HAPs in these EJ and non-EJ communities, as well as the burning behavior of residents in these communities.

## 2. Methods

### 2.1. Study Design and Site Selection

Ambient air was monitored in three EJ and three non-EJ communities. The EJ communities were determined using the EPA’s EJ Screen tool [34]. In EJ Screen, a demographic index is combined with an environmental index to determine an EJ Index for each US census block, yielding a relative rank for each census block in the United States. The demographic index uses the average of an income level factor and a racial minority factor. Specifically, income level is the number or percent of a block group’s population in households where the household income is less than or equal to twice the federal poverty level, and the minority factor is the percent of individuals in a census block who list their racial status as a race other than white alone and/or list their ethnicity as Hispanic or Latino. The demographic indexes count each indicator as adding to the overall potential susceptibility of the population in a block group and assumes that the demographic indicators have equal and additive impacts. The PM_2.5_ was used as the environmental index which in EJ Screen is a fusion of annual average monitored data and modelled data. The PM_2.5_ concentration in each census block is estimated by the EPA’s Office of Research and Development using a Bayesian space–time downscaling fusion model approach [34,35]. Finally, the EJ index for each census block is determined via the product of three terms multiplied together:EJ Index = (environmental indicator) × (demographic index for block group − demographic index for US) × (population for block group)(1)

This results in a rank of the EJ Index for each US census block relative to all other census blocks. A higher EJ index indicates more potential for exposure/risk/proximity to certain facilities and/or a higher percentage minority population. Here, we identified communities with an EJ Index for PM_2.5_ in the highest 10% nationally as EJ locations, and those in the lowest 50% EJ Index as non-EJ locations; see Figure 1 for a map of the area, communities, and monitoring locations. Two existing permanent monitoring sites located in non-EJ communities were used at Del Paso Manor and T Street. To facilitate comparison between EJ and non-EJ communities, two EJ communities with similar characteristics near Del Paso Manor and T Street were identified. The Arden community (EJ) was paired with Del Paso Manor (non-EJ), less than 2 miles away, while the South Natomas community (EJ) was paired with T Street (non-EJ), less than 5 miles away. South Sacramento was selected as an additional EJ community, since there is limited air quality monitoring in the area and multiple census blocks in the community ranked in the top 10% of the national EJ Index for PM_2.5_. The nearby community (within 5 miles) of Colonial Heights was selected as the non-EJ community to pair with the South Sacramento community. Sacramento Metropolitan Air Quality Management District (SMAQMD) then conducted public outreach through local community groups, neighborhood associations, and its Board Members and staff in a search of volunteers to host monitoring equipment.

The HAPs and BC were measured at one site in each of the six communities. Quartz fiber filter samples for quantifying levoglucosan were collected at Del Paso Manor, T Street, and South Sacramento. PM was measured at all sites using low-cost sensors; details can be found in Mukherjee et al. [36]. Details of the sampling and analytical methods are provided in subsequent sections and are summarized in Table 1. In addition to ambient air monitoring, a phone survey was conducted to characterize what indoor burning devices are used and how often people burn with these devices in the EJ and non-EJ communities where the ambient sampling occurred. Data from the ambient measurements and survey were then compared; methods for this comparison are discussed after the discussion of air sampling and analytical methods.

### 2.2. Hazardous Air Pollutants Sampling and Analytical Methods

The HAPs were collected at a site in each community using five-liter Summa stainless-steel canisters at each site. Samples were collected for four sequential 12 h durations starting at either 06:00 or 18:00 PST on days when forecasts indicated that PM_2.5_ concentrations would be high on a particular day. Daytime (06:00 or 18:00) and nighttime (18:00 to 06:00) samples were taken, since mobile source emissions are relatively higher in daytime, and residential wood burning emissions are higher in nighttime [25,26,37]. Thus, a comparison of daytime to nighttime concentrations can show which source type contributes more to ambient HAPs. The sample periods were 19–23 December, 29–30 December, 15–17 January, and 27–30 January. Ten collocated samples were collected at T Street. A total of either 23 or 24 samples were collected at each site, except for at T Street, where 12 samples were collected, 10 of which had collocated samples taken from side-by-side monitors.

Canisters were shipped to Eastern Research Group (ERG) for analysis by EPA Method TO-15 [38] using a gas chromatography-mass spectrometry (GC-MS) system. Gaseous target compounds included 1,3-butadiene, 2,2,4-trimethylpentane (also known as iso-octane), acetonitrile, acetylene, acrolein, acrylonitrile, benzene, carbon tetrachloride, ethylbenzene, m- and p-xylenes, and toluene. These key compounds are among the highest contributors to cancer risk and hazard nationally [39,40]. In addition, many of these compounds are found in wood smoke [7,13,41] and were targeted to examine the potential impact of wood smoke on local concentrations of air toxics. 2,2,4-Trimethylpentane was used as the tracer for fossil fuel combustion, as it is not found in wood smoke [42]. Method detection limits (MDLs) were unique to each canister. Each sample concentration was compared to the sample-specific MDL. Acrylonitrile was never detected in any samples at those detection limits, while all other target compounds were above detection limits in more than 95% of samples. Detection limits were typically below 30 parts per trillion (ppt) for most target HAP compounds. The HAP data were assessed to examine differences in concentrations spatially and temporally, and the HAP data were also compared to levoglucosan, PM_2.5_, and BC data to assess whether there were any covariates that would indicate similar emissions sources.

### 2.3. Filter Collection and Laboratory Analysis

Quartz fiber filters (47 mm, Pall Life Sciences) were collected at Del Paso Manor, T Street, and a South Sacramento site on a forecast basis, generally at the same time as the nighttime HAP samples (i.e., 18:00 to 06:00 PST). Filters were analyzed for organic carbon (OC), elemental carbon (EC), and levoglucosan. Two daytime (06:00 to 18:00 PST) filters were also collected at each of these three sites during January 2017 to evaluate the difference between daytime and nighttime wood smoke concentrations, since residential wood smoke emissions generally occur in the evening when ambient temperatures begin to decrease. At the Del Paso Manor and T Street sites, Thermo Scientific 2025i sequential filtration samplers were used, with a sampling rate of 16.7 L/min. At the South Sacramento site, an AirMetrics mini-vol sampler was used, with a sampling rate of 5 L/min. Sample flow was evaluated before and after each sampling event. Five collocated filters were collected at South Sacramento to evaluate precision. Prior to sampling, filters were pre-baked for 12 h at 500 °C to remove residual organic compounds.

Levoglucosan was determined using a Dionex DX-500 series ion chromatograph with detection via an ED-50/ED-50A electrochemical cell. This cell includes two electrodes: a pH-Ag/AgCl (silver/silver chloride) reference electrode and a “standard” gold working electrode. For the separation, a sodium hydroxide gradient and a Dionex CarboPac PA-1 column (4 × 250 mm) were used. The complete run time was 59 min with an injection volume of 100 µL. More details on the method can be found in Sullivan et al. [43,44,45].

### 2.4. Aethalometer Operations and Calculations

Magee Scientific model AE33 Aethalometers were deployed at a site in each community to measure BC. The AE33 collects ambient aerosol onto a filter tape, and then every five minutes measures the absorbance of the material deposited on the tape at 7 wavelengths, ranging from 370 to 960 nm. The BC concentration is from the 880 nm channel. While BC absorbs uniformly across all wavelengths at which absorbance is measured, brown carbon, which is produced by combustion of wood or other biomass, absorbs primarily in the lower wavelengths of the measured range [46,47]. A commonly used method is to assume that BC is only from wood burning and fossil fuel combustion, so that the difference in absorbance at the 470 nm and 960 nm wavelengths can be used to calculate the quantity of BC_wb_ and BC_ff_. This method assumes that there are no other sources of BC such as coal dust. In the Sacramento emissions inventory, there is little to no impact from coal combustion emissions, so mobile sources and wood burning emissions are the main sources of BC.

To calculate BC_wb_ and BC_ff_, the model described by Sandradewi et al. [48] and summarized in the AE33 User Manual [49] was used. The method uses the concentration values calculated using the 470 nm and 960 nm wavelengths to estimate the percentage of BC from wood burning. Concentrations of BC_wb_ and BC_ff_ are then calculated by multiplying the BC value from the 880 nm channel by the percentages obtained in the previous step. The calculation is based on a model in which the total BC can be divided into pure black and pure brown carbon using the spectral dependence of the absorbing properties of each material. The spectral dependence is described by the Angstrom exponent (α). From the Beer–Lambert Law, the following equations can be obtained: (2)$$\frac{b_{abs}{\left(470\;nm\right)}_{ff}}{b_{abs}{\left(950\;nm\right)}_{ff}}={\left(\frac{470}{950}\right)}^{-\alpha_{ff}}$$
(3)$$\frac{b_{abs}{\left(470\;nm\right)}_{wb}}{b_{abs}{\left(950\;nm\right)}_{wb}}={\left(\frac{470}{950}\right)}^{-\alpha_{wb}}$$
(4)$$b_{abs}\left(\lambda \right)=b_{abs}{\left(\lambda \right)}_{ff}+b_{abs}{\left(\lambda \right)}_{wb}$$
where b_abs_(λ)*_x_* is the absorption coefficient for the BC type *x* (wb or ff) at wavelength λ (470 or 950).

The Sandradewi method requires input values for the Angstrom exponents for pure black and pure brown carbon. For this study, an Angstrom component of 1 for the pure black carbon and a value of 2 for the pure brown carbon were used. These are the default values used by the Aethalometer [23,49] and have been widely used in the scientific literature [25]. Harrison et al. [50] points out that use of the Aethalometer model can have limitations, notably from uncertainty in the Angstrom exponent selected for wood smoke, and when other sources of BC aerosol impact a monitor, such as coal emissions; however, in Sacramento there are no other significant sources of BC except biomass burning and fossil fuel combustion. Th BC_wb_ measurements were compared to levoglucosan to assess the quality of the BC_wb_ calculation.

### 2.5. Routine PM_2.5_ and Meteorological Measurements at Del Paso Manor and T Street

Hourly PM_2.5_ mass and meteorological data were obtained from EPA’s air quality system (AQS) for the pre-existing routine monitoring sites at Del Paso Manor and T Street. At both sites, a MetOne beta attenuation monitor (BAM) 1020 collected hourly PM_2.5_ data, and an R&P 2025 sequential filter sampler was operated as the 24 h Federal Reference Method (FRM) for PM_2.5_ mass. The FRM filters were collected daily, and PM_2.5_ mass was determined by gravimetric analysis. Temperature, RH, wind speed, and wind direction data were collected at both sites using R.M. Young instrumentation. Dew point was calculated from temperature and RH measurements. Instrumentation and operations at these sites followed national guidance set by EPA on operations and methods. Wind and temperature data were used for the analysis of BC data. Each site with Aethalometer and canister measurements was assigned meteorological data from the weather station closest to that monitoring site. The Arden and Del Paso Manor communities were closest to the Del Paso Manor weather station, while the T Street, South Natomas, South Sacramento, and Colonial Heights communities were closest to the T Street weather station.

### 2.6. Data Analysis Methods

#### 2.6.1. Statistical Methods

Statistical methods used here included conditional bivariate probability function plots [51,52], notched box whisker plots, coefficient of divergence calculations, ratio:ratio plots, and Kruskal–Wallis rank sum tests. Unless otherwise noted, the term “significant” is used when the *p*-value of a Student’s *t*-test is less than 0.05, i.e., significant at a 95% confidence level.

To determine whether significant differences in measured concentrations of BC, BC_ff_, and BC_wb_ existed across communities, a Kruskal–Wallis rank sum test was performed on HAP concentrations among the communities. The Kruskal–Wallis test was selected to assess how similar the distribution of data was among sites. Post-hoc Nemenyi tests were applied for pairwise comparisons of BC data between each community to determine which communities’ measurements differed significantly from each other; this test is useful for determining whether multiple groups of data are similar or not when there is a large amount of data [53].

To identify potential sources of black carbon, a conditional bivariate probability function (CBPF) analysis [52] was used. A CBPF analysis is similar to conditional probability function (CPF) analysis which is commonly used to identify emission sources. In CPF analysis, wind direction sectors that have a high probability of pollutant concentrations greater than a selected percentile of the total data are identified. The CBPF analysis enhances this method by incorporating wind speed bins and wind direction. Specifically, where *m* is the number of samples with a concentration greater than the specified percentile, *n* is the total number of samples, *θ* is a wind direction sector, and *j* is the wind speed interval, the following equation is used to determine CBPF.
$$CBPF=\frac{m_{\theta ,j}}{n_{\theta ,j}}$$

The CBPF is able to determine whether certain combinations of wind speed and wind direction have a high probability of producing concentrations above the specified percentile of data. By providing a greater degree of granularity to the analysis, CBPF facilitates the identification of sources that might not be detected by traditional CPF and can also be used to evaluate the characteristics of a source.

The polarPlot function available in the openair R package [51,54] was used to perform CBPF analysis. The CBPF analyses were done for each site where an AE33 was deployed to collect BC, BC_wb_, and BC_ff_ concentrations based on the 80th percentile of the concentrations of each pollutant. The calculations, including the 80th percentile threshold, were performed individually for each site, and radial plots displaying the probability of exceeding the 80th percentile relative to wind speed and direction were created.

Ratio:ratio plots were used to assess how individual species relationships changed as a function of known tracers of wood smoke (BC_wb_) and fossil fuel (2,2,4-trimethylpentane) combustion [42]. In a ratio:ratio plot, the ratio of two given species to another species is plotted on both the *x*- and *y*-axis, e.g., benzene/2,2,4-trimethylpentane versus toluene/2,2,4-trimethylpentane. We examined all possible combinations of the 12 h HAPs relative to the wood smoke indicator BC_wb_ or the mobile source indicator 2,2,4-trimethylpentane to determine under which indicator species the HAPs varied the most. Differences in how ratios change depending on the indicator species in the denominator can indicate which source (i.e., wood smoke or mobile sources) most affects the species in the numerator [55].

#### 2.6.2. Filter Data Analysis

Multiple studies have examined how to best convert levoglucosan to total wood burning PM. Puxbaum et al. [56] provided a review from numerous laboratory tests and suggested using a factor of 7.35 to convert levoglucosan concentration to wood burning organic carbon concentrations for wood burning stoves in the United States based on source profiles from Fine [57]. This is similar to what other studies have used [58,59,60] and is within the typical range used elsewhere, e.g., 10 in Szidat et al. [61] and 11 in Fuller et al. [62]. In an extensive experiment in London, England, Crilley et al. [25] compared multiple conversion factors and found that the Puxbaum et al. [56] factor of 7.35 gave the best agreement compared to a radiocarbon approach. Therefore, the Puxbaum factor was used to determine wood burning PM concentrations as equal to 7.35 × levoglucosan.

For each monitor site, a linear regression between BC_wb_ and levoglucosan was calculated. To make this calculation, hourly BC_wb_ measurements over the 12 h period when filter measurements were collected were averaged together. A y-intercept of 0 was assumed, since BC_wb_ and levoglucosan are both from the same source and should thus trend together to zero [29,62]. The Pearson correlation coefficient for BC_wb_ and levoglucosan was calculated at each site. The site with the highest correlation coefficient between BC_wb_ and levoglucosan was used to develop the relationship between BC_wb_ and wood smoke PM_2.5_.

### 2.7. Phone Survey Methods

During the study period of December 2016 through January 2017, Meta Research, Inc., conducted a phone survey of residents in the six communities where monitoring was conducted. The objectives of the survey were to: (1) assess wood burning behavior; (2) evaluate wood burning activity by type of device used to burn; and (3) compare wood burning activity between EJ and non-EJ communities. Telephone interviews were completed with 900 (444 EJ; 456 non-EJ) Sacramento County residents who owned a wood or pellet burning device (other than an outdoor barbecue) either inside or outside their home. Addresses in each community were matched with listed landline and mobile phone numbers, then selected at random for interviewing.

The margin of error associated with a sample of 900 completed interviews is ±3.3% at the 95% confidence level. That is, there is a 95% chance that the true population parameters lie within 3.3% of the sample statistics. For example, if a response category to a question was chosen by 50% of sample respondents, there is a 95% chance that if the entire county population were surveyed, that same response category would be selected by 46.7–53.3% of all residents (50% ± 3.3%). For the EJ communities, with 444 completed interviews, the margin of error at the 95% confidence level is ±4.7%, while that for the non-EJ communities, with 456 completed interviews, is ±4.6%.

Most of the questions were asked in a closed-ended format which were categorized for quantitative analysis; responses to open-ended questions were not used in quantitative analysis. Interviews took approximately 8 min on average to administer. Respondents were screened for age (18+) and ownership of a wood or pellet burning device, and to confirm residency in Sacramento County. Interviewing took place from 2 December through 19 December 2016, and from 6 January through 22 January 2017. Surveys were not conducted on holidays because response rates are typically lower on those days. A full list of the survey questions is provided in the Appendix A. From these questions, the types of burning devices used in each community, the fraction of homes with EPA-certified devices, the frequency of use of burning devices, the relative number of burn days, and basic demographic information were determined. The relative number of burn days was calculated by how often each respondent used a device, e.g., if a respondent used the device one day per week, the relative number of burn days per week would be 1.

## 3. Results and Discussion

### 3.1. HAPs and Levoglucosan Precision

The precision of HAP measurements was determined using ten collocated (i.e., same place, time, and method) samples at T Street. Precision was calculated using the following formula, where A and B are the concentrations from the two collocated samples.
Precision (%) = 200 × [|(A − B)|/(A + B)](5)

The National Air Toxics Trends Station’s (NATTS) goal for high-precision data is ±15% [63]. As shown in Table 2, median precision was lower than 15% for all pollutants except acrolein (40%) and acrylonitrile (below detection). This information indicates that precision is sufficient to assess spatial variations of >15% for all pollutants except acrolein. Acrolein is known to have potential analytical issues based on canister sampling work done by Eastern Research Group Inc. (ERG) and EPA [64]. Levoglucosan precision was assessed in a similar way using five collocated filter samples collected at South Sacramento. The average precision was 4%, and the *R*^2^ value of the collocated measurements when plotted on a scatter plot was 0.99, indicating high precision.

### 3.2. Comparison of BC_wb_ and Levoglucosan

Levoglucosan is a unique tracer for wood burning. It is typically collected via multi-hour filter measurements, so it cannot be measured continuously; significant labor and equipment are required to collect and chemically analyze the filters. The Aethalometer provides a calculated value of wood burning on an hourly basis and can run with little maintenance at multiple sites but requires validation that the wood burning calculation is correct. With both filter and Aethalometer measurements collected at the Del Paso Manor, South Sacramento, and T Street sites, we assessed how Aethalometer BC_wb_ compared to PM_2.5_ from wood burning estimated from levoglucosan measurements. If BC_wb_ has a strong relationship with levoglucosan, it can be used to determine wood burning PM concentrations from Aethalometer measurements. Levoglucosan was compared to BC_wb_ concentrations when the filters were collected as shown in Figure 2. Correlations among the concentrations of the two species were generally high. At the Del Paso Manor site, which had the highest levoglucosan and therefore wood smoke concentrations, there was a very high correlation between levoglucosan and BC_wb_ (*R*^2^ = 0.95). There were moderate correlations between BC_wb_ and levoglucosan concentrations at the South Sacramento (*R*^2^ = 0.68) and T Street (*R*^2^ = 0.80) sites.

### 3.3. Levoglucosan and Wood Burning PM Estimation

Levoglucosan was used to estimate wood burning PM_2.5_ at the three sites where filters were collected. Filter results are summarized by site in Table 3. At each site, wood burning PM was calculated as described in Section 2. Figure 3 shows the amount of wood burning PM at each site by sample. At the Del Paso Manor and T Street sites, the fraction of PM_2.5_ from wood burning was also calculated using hourly PM_2.5_ measurements. Median levoglucosan concentrations across sites ranged from 0.86 μg/m^3^ at T Street to 1.84 μg/m^3^ at Del Paso Manor; the highest measured levoglucosan concentration was 5.8 μg/m^3^ at Del Paso Manor on the night of 19 December, when PM_2.5_ averaged 69 μg/m^3^ during the 12 h of sampling. Overall, levoglucosan and wood burning PM concentrations were highest at the Del Paso Manor site; concentrations at T Street and South Sacramento sites were similar. At the Del Paso Manor site, 39% of the PM_2.5_ (median value) during the filter sampling periods was from wood burning. At the T Street site, wood burning accounted for 29% of the PM concentrations (median value). At the South Sacramento site, the median concentration of wood burning PM_2.5_ was similar to that at the T Street site (6.8 μg/m^3^ at the South Sacramento site and 6.3 μg/m^3^ at the T Street site), and about half of the median concentration at the Del Paso Manor (13.5 μg/m^3^) site. These results suggest that wood burning emissions are higher around the Del Paso Manor site than around the other sites but are still responsible for a significant amount of the PM_2.5_ at T Street and South Sacramento.

Nighttime concentrations were compared to two daytime samples collected at each site. As discussed later in the results of the phone survey, residential burning occurs predominantly in the evening or nighttime, not during the day, so levoglucosan found during the daytime is likely carried over from the prior night. On 28 and 29 January 2017, samples were collected at 06:00 and 18:00 PST at each site to provide a rough comparison of daytime concentrations to nighttime concentrations. Figure 4 shows the average of burning PM concentrations for the two daytime samples compared to two nighttime samples collected on 27, 28, and 29 January at the Del Paso Manor and T Street sites and on 15 and 16 January at the South Sacramento site. Del Paso Manor and T Street both had sequential samplers, enabling both nighttime and daytime filters to be collected, while South Sacramento had mini-vol samplers that collect only one filter at a time. Daytime samples were not collected at South Sacramento on 27–29 January because nighttime samples were collected. Nighttime concentrations of burning PM were four times higher than daytime concentrations at Del Paso Manor, roughly twice as high as daytime concentrations at the T Street site and roughly 50% higher than daytime concentrations at South Sacramento. Daytime concentrations of burning PM were similar at all three sites. While based on only a handful of samples, these results suggest that wood burning PM_2.5_ is well distributed across these three sites during the daytime and is elevated around Del Paso Manor during the nighttime.

### 3.4. HAPs and BC Concentrations by Community

Concentrations of 2,2,4-trimethylpentane, acrolein, benzene, ethylbenzene, m- and p-xylenes, and toluene were significantly higher (*p* < 0.05) at EJ communities compared to non-EJ communities. For example, benzene concentrations were 0.65 µg/m^3^ in EJ communities and 0.58 µg/m^3^ in non-EJ communities, on average. There was no significant difference between individual EJ/non-EJ site pairs for any species (Table 4), likely due to the smaller overall sample size (*N* = 23 at all locations except T Street which had *N* = 12) when comparing individual site locations. The HAPs that were significantly higher in EJ communities were typically from mobile sources, including 2,2,4-trimethylpentane, an indicator for mobile sources [42]. These results are consistent with EJ Screen which shows the three EJ communities in this study having EJ indices for PM_2.5_ in the highest (worst) decile of the entire United States, predominantly due to the fact of mobile source emissions.

The two likely sources of HAPs in this study—mobile sources and wood burning—peak at different times of a typical day; mobile source emissions peak in the morning during the commute hours with a secondary peak in the evening, while wood burning emissions peak in the evening hours and slowly decline into the early morning hours of the following day. Thus, having samples in both daytime and nighttime in the context of the diurnal BC, BC_wb_, and BC_ff_ patterns allow for a comparison of whether HAPs are predominantly from mobile sources or wood burning; day/night changes in the stability of the atmosphere will also play a role in concentrations. For example, if HAP concentrations are higher in the daytime, mobile sources can be assumed to be a more important source of those HAPs than residential wood burning which predominantly occurs at night. The HAP samples were typically collected during four sequential 12 h daytime and nighttime periods, e.g., 12 h samples were taken starting at 18:00 PST, then at 06:00 PST and 18:00 PST the following day, and at 06:00 PST the day after that.

First, the day/night trends of carbon tetrachloride (CCl_4_), a very long-lived HAP that is no longer emitted in the US, were examined. It was phased out as part of the Montreal Protocol to ban chlorofluorocarbons (CFCs) to repair the ozone hole; because it has a very long residence time in the atmosphere, it is very homogeneous, and concentrations should be approximately 0.085 ppb everywhere in the United States [39,65]. The spatial and time-of-day variations should be minimal, and any deviations would indicate sampling and analytical imprecision. The samples measured during this study show no systematic bias across sites or time of day. This indicates high-quality sampling and analysis results.

In contrast to carbon tetrachloride, most of the HAPs showed large differences between daytime and nighttime concentrations. Statistical tests of daytime/nighttime differences are summarized in Appendix A. Most sites measured higher concentrations in the daytime than in the nighttime for all species except carbon tetrachloride. Daytime concentrations of most HAPs were consistently higher than nighttime concentrations at the Arden, Del Paso Manor, South Natomas, and South Sacramento sites. Higher concentrations in the daytime indicate that the dominant sources of HAPs are more likely mobile source emissions than wood smoke emissions, consistent with the results shown earlier that HAPs have no statistically significant relationship with levoglucosan.

### 3.5. BC Concentrations by Community

The BC concentrations were compared among community sites to assess whether EJ or non-EJ sites measured higher concentrations. The BC concentrations by site are shown in Figure 5 and are also separated out by BC_ff_ and BC_wb_. Overall, BC and BC_wb_ concentrations were significantly higher at the Del Paso Manor site when compared to the other sites, while BC_ff_ was found to be generally similar across all sites (as shown in Appendix A). Based on Mann–Whitney U-tests, BC and BC_wb_ concentrations were higher at non-EJ sites compared to EJ sites at the 95% confidence level. The BC_ff_ concentrations were not significantly different between the EJ and non-EJ sites. Significant site-to-site variations were observed: BC concentrations at the South Natomas EJ site were significantly higher than the nearby non-EJ T Street site, and BC concentrations at the Colonial Heights and Del Paso Manor non-EJ sites were significantly higher than the South Sacramento and Arden EJ sites.

The diurnal and day-of-week variations in BC, BC_wb_, and BC_ff_ were examined to investigate the temporal patterns of concentrations of these species (see Appendix A and Figure 6). Fossil fuel sources are expected be highest during the morning and evening commutes, and wood burning is expected to be highest during the evening. Day-of-week patterns may also exist, as there is typically less driving on Sundays compared to weekdays [66,67,68,69], and biomass burning indicator concentrations are often higher on weekends when residents tend to burn more [2,70,71,72,73].

In Sacramento, the T Street site has a typical urban diurnal pattern with elevated morning BC, while the other sites have a nighttime peak in BC that is higher than the BC morning peak. For all sites, BC_wb_ was much higher at night than during the morning or daytime, consistent with expected wood burning patterns and phone survey results, discussed in detail later. There was also a large variation in nighttime BC and BC_wb_ concentrations, as seen by the large confidence interval surrounding the median line in Figure 5. Overall, results suggest a strong effect from wood burning in the evening at all sites but T Street. The BC_ff_ had a typical diurnal pattern of morning and evening peaks consistent with morning and evening rush hour traffic. The day-of-week patterns for BC, BC_wb_, and BC_ff_ were unexpected with a large drop in all species concentrations between Wednesday and Thursday. Since only eight weeks of data were collected, variations were likely due to the presence of stochastic events such as rainfall (which would decrease concentrations) rather than consistent robust differences in emissions on Wednesday and Thursday compared to other days.

Hourly BC, BC_wb_, and BC_ff_ measurements were combined with meteorological data to try to identify the likely source of these pollutants. The CBPF was used to identify wind speeds and directions associated with the highest quintile of concentrations. For all sites in this study, the highest quintile of concentrations of BC, BC_ff_, and BC_wb_ occurred at very low wind speeds associated with stable and stagnant conditions (see Appendix A). At the T Street site, there were also high concentrations under high-speed winds from the northwest. However, closer examination revealed that this was caused by just 3 h in early December, during which BC_wb_ concentrations were high and wind speeds were also high. It is unlikely that this result indicates a consistent source of BC_wb_, and it is likely caused by a temporary, aberrant emission event. For BC_ff_, the plots suggest that minor effects from fossil fuel sources exist south–southeast of the Colonial Heights, Del Paso Manor, and T Street sites which is the direction where freeways are located. Otherwise, CBPF analysis simply shows that high concentrations occurred with low wind speeds, rather than pointing to a specific source area or direction.

### 3.6. Inter-Comparisons Among Measurements

The association of individual HAPs with BC_wb_ was investigated using linear regression and ratio–ratio plots; correlation of a given HAP with BC_wb_ would indicate that wood burning is a significant source of the HAP, while ratio–ratio plots can indicate which species may have a common source. The relationship of each HAP with 2,2,4-trimethylpentane was also examined, as 2,2,4-trimethylpentane is an indicator of fossil fuel combustion [42,74,75,76]; all results are presented in Table 5. Consistent across all six sites, BC_wb_ and levoglucosan had little correlation with any HAP (i.e., *R*^2^ < 0.20). This finding indicates that wood burning had little relation to HAPs at any site. As expected, carbon tetrachloride had little correlation with any other HAP (*R*^2^ < 0.12), since its concentration levels are representative of global background concentrations. The HAPs that are typically from fossil fuel combustion (benzene, ethylbenzene, m- and p-xylenes, toluene, 1,3-butadiene, 2,2,4-trimethylpentane) had high correlations among themselves at every site (*R*^2^ > 0.70). This means that all of these species were from the same type of source, i.e., mobile source emissions.

Ratio:ratio plots were used to help understand the relative contribution of wood smoke and fossil fuel combustion to HAPs, OC, EC, and PM_2.5_. The relative importance of the two sources can be found by plotting the ratio of each species relative to tracers of wood smoke (represented by BC_wb_) or fossil fuel combustion (represented by 2,2,4-trimethylpentane). High correlation in a ratio:ratio plot indicates that there is little impact from the applied tracer on the species in question, whereas low correlation indicates a high effect from the tracer. Examples are given in Appendix A, which highlights the benzene/toluene and acetylene/1,3-butadiene results. These figures indicate that there was little to no impact from wood burning on HAP concentrations. When divided by BC_wb_, the HAPs were highly correlated on the ratio:ratio plots, but when divided by 2,2,4-trimethylpentane, the results clustered together tightly. These relationships indicate that a singular source of emissions dominated the species shown in the figures. Similar results were seen for most of the hydrocarbons (benzene, toluene, ethylbenzene, acetylene, m- and p-xylenes, and 1,3-butadiene). However, the oxygenated, nitrogenated, and chlorinated compounds, such as acetonitrile and acrolein, did not display this characteristic relationship when divided by either BC_wb_ or 2,2,4-trimethylpentane, indicating that these compounds were not associated with BC_wb_ or 2,2,4-trimethylpentane or were not measured due to the difficulty in measuring such compounds via canisters.

### 3.7. Phone Survey Results and Comparison to Ambient Measurements

Overall, the survey results show no significant differences between EJ and non-EJ communities in the type of device owned, fraction of homes burning with fireplaces, fraction of homes burning day and night, fraction of homes burning only at night, or fraction of homes with EPA-certified burning devices. The only significant differences are that homes in non-EJ communities burn with an indoor device more often than homes in EJ communities do and, thus, have a higher relative number of burn days than EJ communities. Specifically, the most commonly owned device was an indoor fireplace, owned by 79–80% of all respondents in non-EJ and EJ communities; other devices, such as pellet stoves, accounted for less than 10% of responses. Those respondents who do burn with any device do so between one and two days per week. This frequency increases with the more devices a respondent owns. Of all respondents owning an indoor fireplace, those in non-EJ communities burn significantly more than those in EJ communities (0.35 and 0.23 burn days per week, respectively). Among respondents who burn with fireplaces, those in non-EJ communities are burning significantly more often than respondents in EJ communities (1.83 and 1.20 days per week, respectively).

Table 6 summarizes the survey results and ambient HAPs and BC_wb_ results. Non-EJ communities have more burning device usage than EJ communities and, thus, a significantly higher relative number of burn days. This difference in the number of burn days likely means there are more emissions from wood burning in non-EJ communities which corresponds with the significantly higher observed BC_wb_ in non-EJ communities. However, HAPs have an opposite relationship; for example, in the EJ communities where less wood burning occurs than in non-EJ communities, concentrations of multiple HAPs are higher. These results support all other analyses of the ambient data showing that wood burning has little impact on ambient HAP concentrations.

## 4. Conclusions

In this first detailed community monitoring study during the winter in Sacramento County, we found that concentrations of six HAPs (benzene, toluene, ethylbenzene, m- and p-xylenes, 2,2,4-trimethylpentane, and acrolein) and BC_ff_ were significantly higher at EJ communities than at non-EJ communities. Even though wood smoke emissions can contribute to HAPs, results from this study consistently showed that wood burning has little influence on the ambient levels of HAPs and that fossil fuel combustion was the main source of HAPs. The BC_wb_ had a very high correlation with collocated measurements of the wood burning tracer levoglucosan (*R*^2^ across three sites of 0.68 to 0.95), on the high end of reported literature values, indicating that BC_wb_ was a robust indicator for wood burning in Sacramento. Thus, Aethalometer BC_wb_ data could be used from all sites to assess wood burning contributions. Wood burning accounted for 29–39% of the nighttime PM_2.5_ at the two sites with hourly PM_2.5_ and filter measurements (T Street and Del Paso Manor). The BC_wb_ concentrations were significantly higher at non-EJ communities than at EJ communities. As shown in the phone survey results, the higher BC_wb_ concentrations in non-EJ communities were mostly likely due to the more burning with indoor fireplaces in the non-EJ communities. Wood burning was not a significant source of HAPs. The HAPs and BC_ff_ were higher in EJ communities compared to non-EJ communities, indicating mobile source emissions were likely higher in EJ communities.

## Figures and Tables

**Figure 1 ijerph-17-01080-f001:**
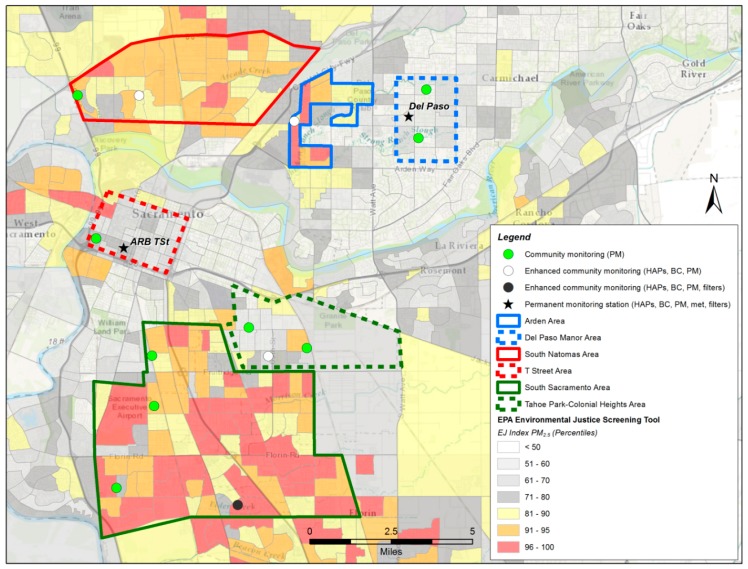
The Sacramento County communities where monitoring was conducted, the monitoring locations and pollutants monitored, and the Environmental Justice Index for PM_2.5_ from EPA’s EJ Screen. HAPs: hazardous air pollutants. BC: black carbon. PM: particulate matter.

**Figure 2 ijerph-17-01080-f002:**
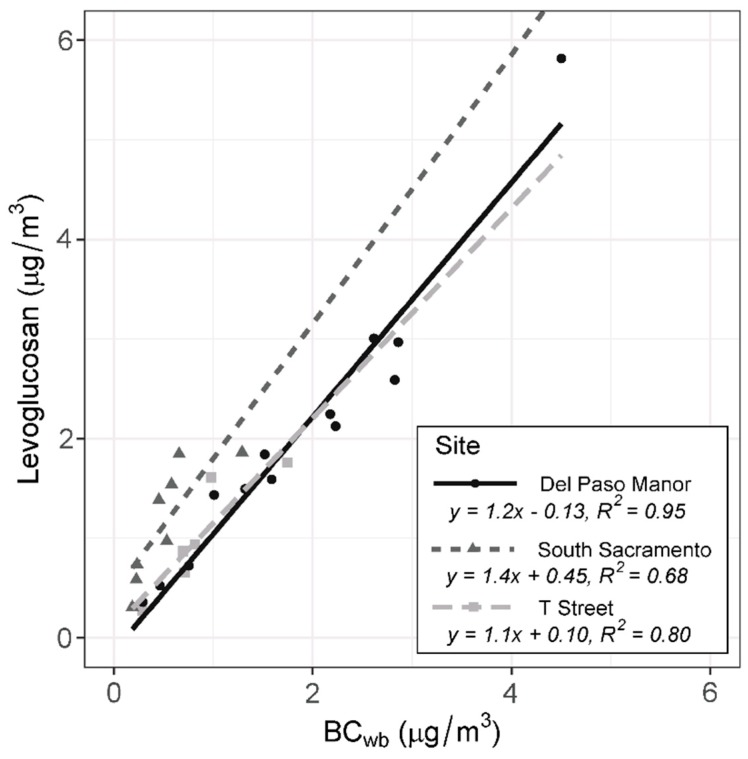
BC_wb_ (μg/m^3^) concentrations compared to levoglucosan (μg/m^3^) at the Del Paso Manor, South Sacramento, and T Street sites.

**Figure 3 ijerph-17-01080-f003:**
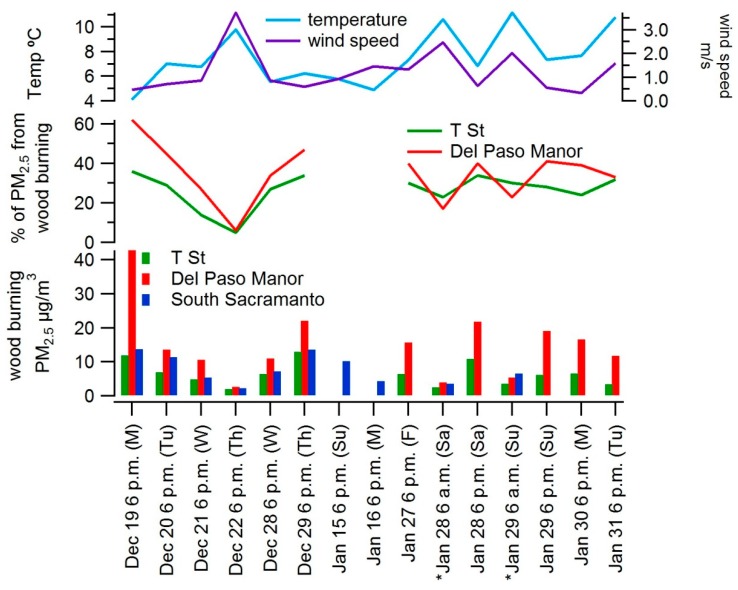
Calculated residential wood burning PM_2.5_ (μg/m^3^) from levoglucosan measurements, percent of PM_2.5_ from residential wood burning and temperature and wind speed for each 12 h period when filters were collected; listed time indicates start time of sample and asterisks indicate the daytime samples. Temperature and wind speed data are shown for the T Street (T St) location. Percent of PM_2.5_ from residential wood burning is based on hourly measurements for PM_2.5_ at T Street and Del Paso Manor.

**Figure 4 ijerph-17-01080-f004:**
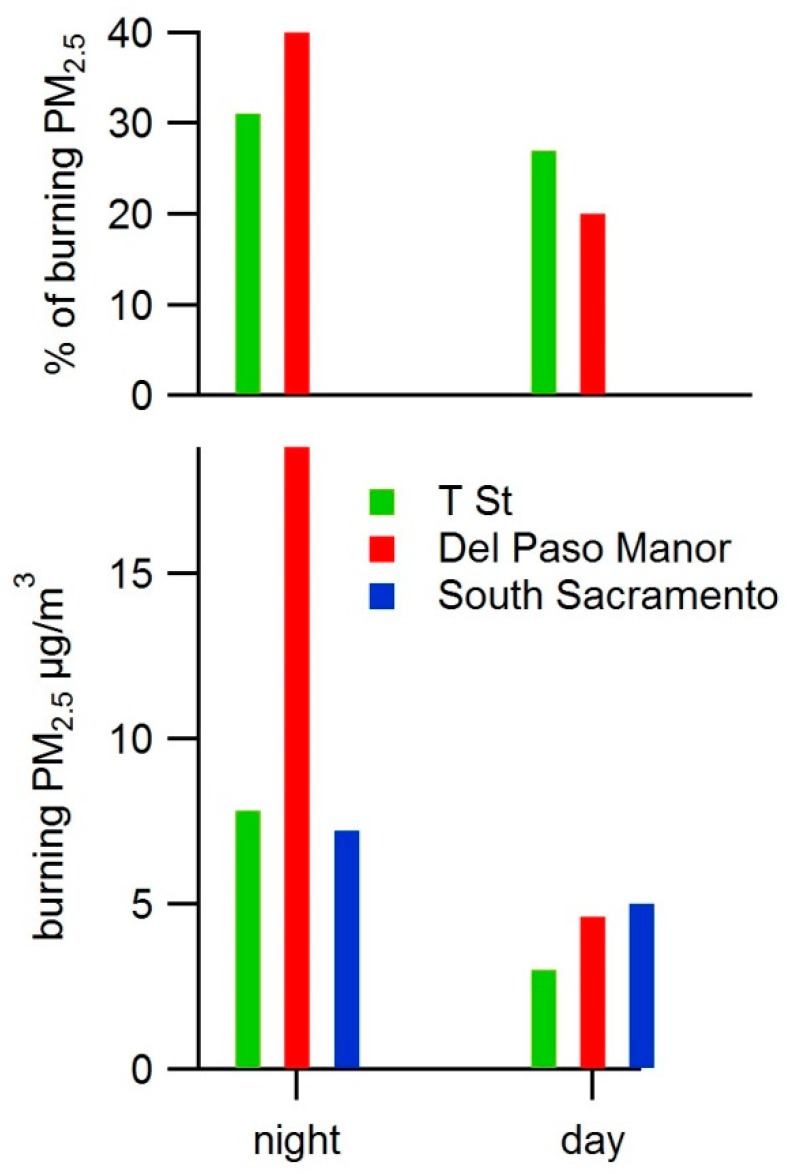
Average fraction of (top figure) and total (bottom figure) residential wood burning PM_2.5_ concentrations (μg/m^3^) during two daytime samples (28 and 29 January 2017 at the Del Paso Manor, T Street, and South Sacramento sites) and two nighttime samples (27 through 29 January 2017 at the Del Paso Manor and T Street sites and 15 and 16 January 2017 at the South Sacramento site).

**Figure 5 ijerph-17-01080-f005:**
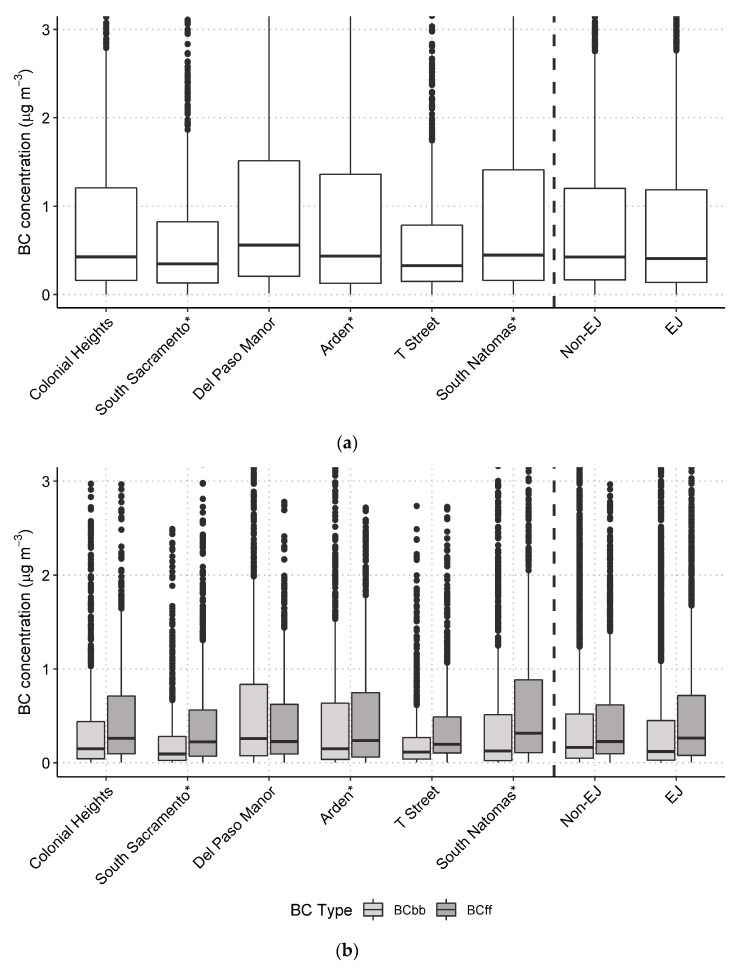
Notched box plot of concentrations by site and for all EJ and all non-EJ sites, from December 2016 through January 2017 for (**a**) BC and (**b**) BC_wb_ and BC_ff_. Data values above 3.1 μg/m^3^ are not shown. * Indicates an EJ site. Boxes are the interquartile range (IQR), and the notch is the median and 95% confidence interval about the median; the whiskers go to 1.5 × IQR, and data beyond this are shown as individual points.

**Figure 6 ijerph-17-01080-f006:**
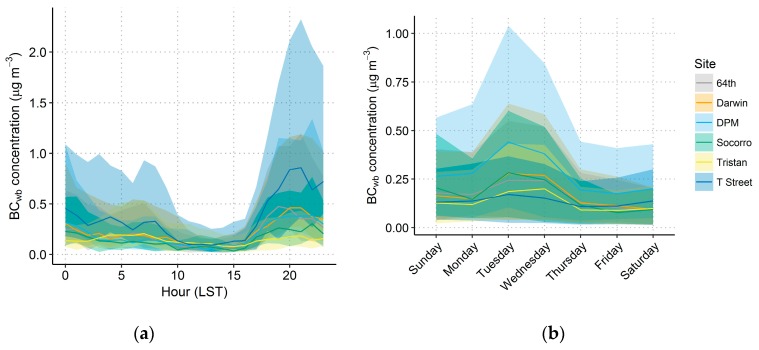
Diurnal (**a**) and day-of-week (**b**) plots for BC_wb_ by site; the median is shown as a line, and the shading indicates the 95th confidence interval around the median. The 95th confidence interval was calculated based on the rank-ordered data values obtained from the binomial quantile function at probabilities of 0.025 and 0.975. DPM: Del Paso Manor.

**Table 1 ijerph-17-01080-t001:** Measurements conducted during the study period of 1 December 2016 through 31 January 2017.

Pollutant	Method	Resolution	Frequency	Colonial Heights (Non-EJ)	South Sacramento (EJ)	T Street (Non-EJ)	South Natomas (EJ)	Del Paso Manor (Non-EJ)	Arden (EJ)
BC, BC_ff_, BC_wb_	Aethalometer	Hourly	Continuous	X	X	X	X	X	X
PM_2.5_	AirBeam sensor	1 min	Continuous	X	X	X	X	X	X
PM_2.5_	Met One BAM	Hourly	Continuous			X		X	
PM_2.5_	FRM R&P 2025	24 h	Daily			X		X	
Gaseous HAPs	Canister sample; TO-15	12 h	Episodic	X	X	X	X	X	X
Levoglucosan, OC, EC	Filter collected with mini-vol or Thermo 2025i filter sampler	12 h	Episodic, coincident with canister samples		X	X		X	
Wind speed, direction, temperature	R.M. Young company (ultrasonic)	Hourly	Continuous			X		X	

EJ: Environmental Justice community. BAM: beta-attenuation monitor. FRM: federal reference method. R&P: Rupprecht & Patashnick Co., Inc. OC: organic carbon. EC: elemental carbon.

**Table 2 ijerph-17-01080-t002:** Collocated HAPs and levoglucosan sample results (μg/m^3^).

Analyte	Emissions Sources	*N* Collocated Samples	Median Precision %	Average Precision %	Average Concentration
2,2,4-Trimethylpentane (iso-octane)	Mobile source tracer	10	5.1	5.9	0.3
Acetylene	Mobile source tracer	10	5.3	6.2	2.96
Benzene	Air toxic	10	6.3	8.5	0.62
Carbon Tetrachloride	Air toxic; internal QC	10	5.8	8.5	0.09
Toluene	Combustion indicator	10	11.0	12.6	1.9
m- and p-Xylenes	Combustion indicator	10	9.4	14.0	0.59
Ethylbenzene	Air toxic	10	9.0	15.2	0.19
Acetonitrile	Air toxic	10	12.7	17.2	0.14
1,3-Butadiene	Air toxic	10	12.1	18.2	0.17
Acrolein	Air toxic	10	40.0	46.0	0.27
Acrylonitrile	Air toxic	10	<MDL	<MDL	<MDL
Levoglucosan	Wood burning	5	3.3%	4.4%	1.1

MDL: method detection limit.

**Table 3 ijerph-17-01080-t003:** Summary of filter results (μg/m^3^) by site; Del Paso Manor and T Street are non-EJ and South Sacramento is an EJ site.

Statistic	Del Paso Manor	T Street	South Sacramento
*N* 12 h samples (*N* night, day)	13 (11, 2)	13 (11, 2)	10 (8, 2)
Median PM_2.5_	39	24	n/a
Median levoglucosan	1.84	0.86	0.93
Median wood burning PM	13.5	6.3	6.8
% PM from wood burning	39%	29%	n/a

**Table 4 ijerph-17-01080-t004:** MDL and site concentrations (μg/m^3^) for each HAP, and p-value of significance between all EJ and non-EJ sites. Bold indicates significant differences at the 95% confidence level used by the Mann–Whitney U-test. Sites are ordered by non-EJ/EJ pair.

Analyte	MDL	Colonial Heights (Non-EJ)	South Sacramento (EJ)	T Street (Non-EJ)	South Natomas (EJ)	Del Paso Manor (Non-EJ)	Arden (EJ)	All Non-EJ	All EJ	*p*-Value (Between All Non-EJ and All EJ Sites)
1,3-Butadiene	0.026	0.166	0.163	0.167	0.222	0.154	0.224	0.160	0.202	0.12
2,2,4-Trimethylpentane	0.010	0.304	0.367	0.285	0.449	0.336	0.467	0.316	0.428	**0.02**
Acetonitrile	0.051	0.127	0.121	0.153	0.161	0.146	0.140	0.141	0.141	0.33
Acetylene	0.029	3.714	3.162	2.905	2.897	2.153	3.079	2.814	3.046	0.21
Acrolein	0.120	0.234	0.290	0.317	0.260	0.235	0.339	0.249	0.296	**<0.01**
Benzene	0.021	0.562	0.593	0.598	0.776	0.585	0.798	0.580	0.722	**0.05**
Carbon Tetrachloride	0.016	0.085	0.088	0.089	0.083	0.090	0.089	0.087	0.087	0.07
Ethylbenzene	0.019	0.176	0.243	0.209	0.244	0.154	0.259	0.171	0.230	**0.02**
m- and p-Xylenes	0.040	0.523	0.564	0.603	0.814	0.478	0.896	0.515	0.758	**<0.01**
Toluene	0.017	0.973	1.289	1.146	1.697	0.973	1.648	1.004	1.544	**<0.01**

**Table 5 ijerph-17-01080-t005:** Matrix of median correlation coefficient (*R*^2^) of HAPs, PM, and filter measurements (where available) across all six monitoring sites. Correlations greater than 0.70 are shown in bold.

Parameters	Acetonitrile	Acetylene	Acrolein	BC	BC_ff_	BC_wb_	Benzene	CCl_4_	Ethylbenzene	m,p-Xylene	Toluene	1,3-Butadiene	iso-octane	Levoglucosan
Acetonitrile	N/A	0.17	0.03	0.10	0.09	0.07	0.47	0.01	0.36	0.33	0.35	0.37	0.36	0.21
Acetylene	0.17	N/A	0.00	0.11	0.05	0.12	0.53	0.12	0.46	0.43	0.43	0.54	0.43	0.07
Acrolein	0.03	0.00	N/A	0.07	0.05	0.06	0.02	0.06	0.00	0.00	0.00	0.02	0.00	0.11
BC	0.10	0.11	0.07	N/A	**0.71**	**0.84**	0.10	0.00	0.10	0.08	0.09	0.08	0.11	**0.85**
BC_ff_	0.09	0.05	0.05	**0.71**	N/A	0.31	0.06	0.01	0.04	0.03	0.03	0.06	0.05	0.05
BC_wb_	0.07	0.12	0.06	**0.84**	0.31	N/A	0.09	0.00	0.11	0.09	0.10	0.06	0.11	**0.88**
Benzene	0.47	0.53	0.02	0.10	0.06	0.09	N/A	0.03	**0.80**	**0.78**	**0.84**	**0.87**	**0.78**	0.17
CCl_4_	0.01	0.12	0.06	0.00	0.01	0.00	0.03	N/A	0.06	0.06	0.06	0.08	0.07	0.00
Ethylbenzene	0.36	0.46	0.00	0.10	0.04	0.11	**0.80**	0.06	N/A	**0.97**	**0.90**	0.67	**0.90**	0.15
m- and p-Xylene	0.33	0.43	0.00	0.08	0.03	0.09	**0.78**	0.06	**0.97**	N/A	**0.85**	0.68	**0.85**	0.13
Toluene	0.35	0.43	0.00	0.09	0.03	0.10	**0.84**	0.06	**0.90**	**0.85**	N/A	**0.72**	**0.91**	0.17
1,3-Butadiene	0.37	0.54	0.02	0.08	0.06	0.06	**0.87**	0.08	0.67	0.68	**0.72**	N/A	0.65	0.11
iso-octane	0.36	0.43	0.00	0.11	0.05	0.11	**0.78**	0.07	**0.90**	**0.85**	**0.91**	0.65	N/A	0.16
Levoglucosan	0.21	0.07	0.11	**0.85**	0.05	**0.88**	0.17	0.00	0.15	0.13	0.17	0.11	0.16	N/A

**Table 6 ijerph-17-01080-t006:** Summary of survey results by EJ and non-EJ area, plus average concentrations of HAPs and BC_wb_ (μg/m^3^) by EJ and non-EJ area. Bold indicates significant differences at the 95% confidence level via the Mann–Whitney U-test; “question” indicates the survey question from which the data are derived (see Appendix A for the list of questions).

Metric	EJ	Non-EJ
Fraction of homes with fireplace (question S3)	80	79
Fraction of homes with any indoor device (question S3)	91	89
Fraction of homes with wood or pellet stove (question S3)	7	8
Fraction of homes with only outdoor burning (question S3)	7	5
Fraction of homes with outdoor and indoor burning (question S3)	92	91
Relative burn days with indoor fireplace (question S3 × 4.0a; × 4.1a)	**0.27**	**0.43**
Relative burn days with wood or pellet stove (question S3 × 4.2a)	0.48	0.7
Relative burn days with any indoor device (question S3 × 4.0,4.1,4.2)	**0.31**	**0.48**
Of households that burn, fraction of homes burning at night with indoor fireplace (question 4.0b)	67	79
Of households that burn, fraction of homes burning day with indoor fireplace (question 4.0b)	10	9
Of households that burn, fraction of homes burning day and night with indoor fireplace (question 4.0b)	19	9
Of households that burn, fraction of homes burning at night with fireplace (question S3 × 4.0b)	76	57
Of households that burn, fraction of homes burning at night with wood or pellet stove (question S3 × 4.0b)	n/a	n/a
Fraction of homes burning with certified device (question 4.0c)	34	28
Fraction of homes burning indoor fireplace (questions 5.2 and 5.4)	**28**	**38**
Fraction of homes burning on “Check Before You Burn” days (question 5.2 and 5.4)	46	47
1,3-Butadiene μg/m^3^	0.18	0.16
2,2,4-Trimethylpentane μg/m^3^	**0.37**	**0.31**
Acetonitrile μg/m^3^	0.14	0.14
Acetylene μg/m^3^	2.99	2.92
Acrolein μg/m^3^	0.27	0.26
Benzene μg/m^3^	**0.65**	**0.58**
Carbon Tetrachloride μg/m^3^	0.09	0.09
Ethylbenzene μg/m^3^	**0.20**	**0.18**
m- and p-Xylenes μg/m^3^	**0.64**	**0.53**
Toluene μg/m^3^	**1.34**	**1.03**
BC_wb_ μg/m^3^	**0.41**	**0.48**

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
