# Peer review of "Assessment of Ambient Air Toxics and Wood Smoke Pollution among Communities in Sacramento County"

_ijerph, 2020, doi:10.3390/ijerph17031080_

Round 1
Reviewer 1 Report
The manuscript is improved and comments from R1 are addressed well.
Reviewer 2 Report
My comments have been considered within the Revision.
This manuscript is a resubmission of an earlier submission. The following is a list of the peer review reports and author responses from that submission.
Round 1
Reviewer 1 Report
Overall: it is a well-written and informative manuscript. I would like to see a comparison of the results from this study with previous and related work. As written, the manuscript is good on reporting the ‘what’ and ‘why’. It will be much stronger if the results are compared and discussed with similar air quality measurements and findings, especially from the EJ/non-EJ perspective on air quality. Specific comments are below:
Abstract:
Please define the problem and state the objectives. Define the 'environmental justice' community. Define BC before the first use. Define ‘ratio-ratio’. Define ‘indoor devices’. In general, the abstract jumps all over. The list of measured pollutants keeps growing with the reading. The scope seems to be changing as well. Some measurements, some phone interviews….it is loosely connected and difficult to follow.Methods:
Section 2.1 – several opening sentences are redundant with the Introduction. Line 107 – define the acronym.Results:
Figure 2 – please show the trendline equations. Table 4? – please correct the numbering. Please also indicate which of the sites are EJ and non-EJ. Define BAM. Table 5 – please check the statistical analysis values. Some of them appear to be too close to call significant. Please report 3 decimal point values for the averages and p values. L519 – define CBPF. L533 – please remind readers of the definition ‘ratio-ratio plots’. L574 – define ‘fireplace inserts’.Conclusions:
L604 – please add a few qualifying words to remind readers what the ‘filters were collected’ means and also the meaning of the two acronyms used.Author Response
Overall: it is a well-written and informative manuscript. I would like to see a comparison of the results from this study with previous and related work. As written, the manuscript is good on reporting the ‘what’ and ‘why’. It will be much stronger if the results are compared and discussed with similar air quality measurements and findings, especially from the EJ/non-EJ perspective on air quality. Specific comments are below:
Abstract:
Please define the problem and state the objectives. Define the 'environmental justice' community. Define BC before the first use. Define ‘ratio-ratio’. Define ‘indoor devices’. In general, the abstract jumps all over. The list of measured pollutants keeps growing with the reading. The scope seems to be changing as well. Some measurements, some phone interviews….it is loosely connected and difficult to follow.
Methods:
Section 2.1 – several opening sentences are redundant with the Introduction. Line 107 – define the acronym.
We have revised the methods to be less redundant, and defined the acronym.
Results:
Figure 2 – please show the trendline equations. Table 4? – please correct the numbering. Please also indicate which of the sites are EJ and non-EJ. Define BAM. Table 5 – please check the statistical analysis values. Some of them appear to be too close to call significant. Please report 3 decimal point values for the averages and p values. L519 – define CBPF. L533 – please remind readers of the definition ‘ratio-ratio plots’. L574 – define ‘fireplace inserts’.
We have:
Added trendline to Figure 2 Updated the Table numbering Updated the table 3 caption to state “Del Paso Manor and T Street are non-EJ and Sough Sacramento is an EJ site” removed “BAM” from the line in the Table as it is covered in the methods and not relevant here Checked the significance tests. The significance is based on the Mann-Whitney U-test, comparing the concentration population of the data collected at EJ and non-EJ sites. Updated the p-values to have 3 decimal point values. We did not do so for the concentrations, since the precision of the analytical method Confirmed that CBPF is defined in the methods at first usage Added text on ratio-ratio plots. Removed the text “insert”Conclusions:
L604 – please add a few qualifying words to remind readers what the ‘filters were collected’ means and also the meaning of the two acronyms used.
We have clarified this statement: Wood burning accounted for 29%-39% of the nighttime PM2.5 at the two sites with hourly PM2.5 and filter measurements (T Street and Del Paso Manor
Reviewer 2 Report
It is a very good paper. The subject is of great relevance, not only for Sacramento but for all regions where Wood is burned for heating processes and People are affected by the Wood Smoke.
I have the following suggestions or Questions for clarification or some improvements.
1. Abstract: There should be no abbreviations in the Abstract.
2. Introduction, line 61,62: The item "environmental justice area or community" is not known in Europe. The meaning should be explained with one sentence: The first Impression is that an environmental justice area is an area with a good Environment. So, without reading the EPA definitions a short explaining (or defining) sentence could be given, e.g.: "EJ is an area with more environmental Problems. Non-EJ could mean less environmental Problems". If this is Right.
Methods:
3. Lines 94 - 98: It is not clear whether a high or a low EJ Index is better. Please, explain it understandably.
4. Line 100: What is "leverage"?
5. Lines 101 f: The map of Figure 1 should be mentioned hereThe names mentioned here in the text should be written into the map of Figure 1. Also the areas should be dirctly named in Figure 1, not in the legend. This figure 1 is difficult to read and difficult to understand. It should be improved as mentioned.
6. Line 110: 13 community sites were identified. For what? What are community sites? Measurement sites? Please, give an explanation for community sites.
7. Line 111/112 and later: What is "T-street" or "T-street community"? Please, mark this T-street in Figure 1. In the moment I cannot see a T-street there.
8. Table 1: This table is confusing (unclear). A much better design would be to write horizontally the sites and vertically the determined compounds. The lines in the columns should be marked where appropriate.
In the table the measurement principle should be mentioned (perhaps under Cellection method).
9. Line 130: In the Headline of 2.2 for HAPs the full word should be written. If somebody reads this headline without reading the explanation for this abbraviation somewhere on the pages before he will not understand what HAPs are.
10. Lines 131 - 144: The Content of this text could be would be much clearer, better understandable and Shorter depicted in a table.
11. Table 2 can be deleted. The MDL dtection limits are included in Table 5 (and give more sense there).
12. Lines 167 and 168 and at other places: The time is given here 18:00 to 06:00 PST. In the lines 133 - 135 it is written 6 a.m. to 6 p.m. The same time terms should be used in the paper (see also line 405, Figure 3 etc.).
13. Line 176: After bakening: Are the filters conditioned under certain humidity and temperature conditions to get comparable results for weighing before and after sampling?
14. Lines 194, 195: What is with diesel soot? Is it considered to be BCff?
15. Line 198ff: 880nm? Should it be 960 nm? In the equations the wavelength is 950 nm, in line 191 it is 960 nm?
16. Lines 202 to 212: Are these equations used for own calculations of BCwb and BCff. Usually these equations are included in the software of the Aethalometer AE33 with a direct displayed output of e.g. BCwb.
17. Line 215: What is SMAQMD, what ARB?
18. Line 219 and Table 1: Is the R.M. Young systems instrumentation an ultrasonic wind instrument to measure the low wind speeds or is it a propeller Instrument? Should be mentioned in Table 1.
19. Line 236: Is PPD60PV-T2 a low cost sensor?
20. Line 239: Usually the manufacturers Default size Distribution has to be adapted to the local optical PM properties by comparison measurements?
21. Line 243: STL?
22. Line 250: FEM or FRM?
23. 2.7.1 Statistical Methods: It would be better to describe the statistical Methods directly there where they are used (at the results). Then it would be better understandable when the application comes directly behind the description of the used statistical method. It avoids also double descriptions. This is also better for the lines 289 to 297.
24. Line 302: There is a fsctor of 7.35 used to convert levoglucosan concentration to Wood Burning organic Carbon concentrations. Later Wood Burning PM is presented (e.g. in line 308 and in in Table 4). Is that calculated with the levoglucosan times the factor 7,35? Or how is Wood Burning PM calculated? See also lines 385/386.
25. Lines 309 to 315. This could be described within the results chapter when the Regression diagrams are shown.
Results:
26. Headline: HAPS … full word!
27. Lines 348 to 359: In the field of Air Quality mesurements we talk About Uncertainties according ISO 20988:2007, not precision.
28. Line 350/351: To calculate the Precision (better uncertainty) A and B are the concentrations from two collected samples. Are these sample collected at the same time and the same place with the same method? With other words Double Determinations?
29. Table 3: The concentrations are depicted here in ppb. They should be given here in µg/m3 according to the result's units in Table 5.
30. Figure 2 is too small and the curves are partially not readable. Why not using the whole print space. DPM > write the full name like for the other two. Write the equations for the regression lines into the diagram.
31. I forgot what is BAM? Please, use a better understandable word for it.
32. Line 414/415: The sentence is based on only 3 day time samples. Please, mention this as limitation for this message in the sentence.
33. Line 419: Table 41? It is Table 4! The first line in the table is confusing: N 12-hr samples are all (total number) of samples. In brackets are the night time sample from these total number of samples. It would be even better to write the little number of daytime samples in brackets.
34. Figure 3: For better understanding the daytime samples should be named above the bars in the diagram. The hours on the x-axis are the starting times? This should be mentioned in the caption.
35. Figure 4: The y-scale is not % of burning PM, it is burning % of PM2.5. In the lower diagram it is also PM2.5.
36. Lines 431 and 432 (Figure caption): ...during two daytime samples .... and two nighttimesamples according Figure 3.
37. Line 440: What is overall sample size?
38. Table 5: Where are the new site names described before? They should be marked and named in Figure 1. The pairs belonging together should be separated with bigger lines from each other.
39. Line 456: Daily patterns? where can we see these?
40. Lines 457 ff: For the difference od daytime and nighttime is has also the stability of the atmosphere to be considered: Nighttime mostly stable, daytime instable (according the wind Speed in Figure 3).
41: Figure 5. For comparison it would be helpful if the bars of BCff and BCwb are directly side by side in one diagram.
42. Line 501: (see Supplemental Information Figures S-1-S-3). This would be good to show here, at least one of these diagrams.
43. Line 505: You can add: ...since People have mor time for heating.
44. 515 to 517: of cours dependent on the stability of the atmosphere, e.g. indicated by wind speed at these days according Fig. 3.
45. Line 519: What is CBPF? It may be explained earlier but I forgot it and cannot find the explanation.
46. Line 522: The figure S-4 from the Supplemental Information would be helpful to be shown here.
47. Line 533: The correlation of HAP with BCwb should be shown.
48. Lines 546, 547 and diagrams of Figure 6. These depicting of correlations are confusing. High correlation indicates little impact from the applied traces on the species in question, whereas correlation indicates high impact ....
Here another way of correlation should be found where high correlation indicates a high impact and low correlation indicates a low impact according common sense.
49. Lines 578/579: Non- EJ communities have more device usage than EJ communities ... significantly higher number of burning days... In the seventh line of table 6 there it is just opposit?
50. Lines 9 to 13 in the table 6: What are Of burners?
51. Conclusions, lines 593 ton 612: This text is more a summary than conclusions. In the conclusions should be explaine why have fossil fuel HAPs no significant differences, why is BCwb higher in Non-EJ communities etc. Such Points are concluded before in the textxs but not concluded here. This chapter should be a summary of conclusions not only a summary of results.
Author Response
It is a very good paper. The subject is of great relevance, not only for Sacramento but for all regions where Wood is burned for heating processes and People are affected by the Wood Smoke.
I have the following suggestions or Questions for clarification or some improvements.
Abstract: There should be no abbreviations in the Abstract.Check style guide
Introduction, line 61,62: The item "environmental justice area or community" is not known in Europe. The meaning should be explained with one sentence: The first Impression is that an environmental justice area is an area with a good Environment. So, without reading the EPA definitions a short explaining (or defining) sentence could be given, e.g.: "EJ is an area with more environmental Problems. Non-EJ could mean less environmental Problems". If this is Right.We have updated the text with EPA’s definition of EJ, and to state that these areas have “higher pollution levels than non-EJ areas”
Methods:
Lines 94 - 98: It is not clear whether a high or a low EJ Index is better. Please, explain it understandably.We have added a sentence to explain what a higher EJ index means: “A higher EJ index indicates more potential for exposure/ risk/ proximity to certain facilities, and/or a higher percentage minority population.”
Line 100: What is "leverage"?We have revised the sentence to be clearer.
Lines 101 f: The map of Figure 1 should be mentioned hereThe names mentioned here in the text should be written into the map of Figure 1. Also the areas should be dirctly named in Figure 1, not in the legend. This figure 1 is difficult to read and difficult to understand. It should be improved as mentioned.We have updated Figure 1 to have the six communities labeled directly on the map, and removed the term “EPA Environmental Justice Screening Tool” from the legend. We have updated the text at line 101 to refer to Figure 1, and made the community names in the figure consistent with the text. The areas are mentioned in lines 101-106, where we discuss each community pair.
Line 110: 13 community sites were identified. For what? What are community sites? Measurement sites? Please, give an explanation for community sites.We have clarified this that HAPs, BC and PM were measured at one site in each community.
Line 111/112 and later: What is "T-street" or "T-street community"? Please, mark this T-street in Figure 1. In the moment I cannot see a T-street there.Figure 1 updated.
Table 1: This table is confusing (unclear). A much better design would be to write horizontally the sites and vertically the determined compounds. The lines in the columns should be marked where appropriate.In the table the measurement principle should be mentioned (perhaps under Cellection method).
We have revised Table 1 as suggested.
Line 130: In the Headline of 2.2 for HAPs the full word should be written. If somebody reads this headline without reading the explanation for this abbraviation somewhere on the pages before he will not understand what HAPs are.We have updated as suggested
Lines 131 - 144: The Content of this text could be would be much clearer, better understandable and Shorter depicted in a table.We have shortened this section to make it clearer, which should then not require a table.
Table 2 can be deleted. The MDL dtection limits are included in Table 5 (and give more sense there).Done
Lines 167 and 168 and at other places: The time is given here 18:00 to 06:00 PST. In the lines 133 - 135 it is written 6 a.m. to 6 p.m. The same time terms should be used in the paper (see also line 405, Figure 3 etc.).We have made this consistent in the text and Figures as 6 a.m. and 6 p.m.
Line 176: After bakening: Are the filters conditioned under certain humidity and temperature conditions to get comparable results for weighing before and after sampling?Filters were not weighted for mass before and after sampling; rather they were baked to remove any organic residue prior to sampling, and then analyzed for OC and EC after sampling.
Lines 194, 195: What is with diesel soot? Is it considered to be BCff?BCff is from all fossil fuel combustion, but here would predominantly be from diesel emissions.
Line 198ff: 880nm? Should it be 960 nm? In the equations the wavelength is 950 nm, in line 191 it is 960 nm?The instrument defines the BC concentration as that measured at 880 nm, and the manual and cited references utilize the optical measurements at other wavelengths to separate out BCff and BCwb.
Lines 202 to 212: Are these equations used for own calculations of BCwb and BCff. Usually these equations are included in the software of the Aethalometer AE33 with a direct displayed output of e.g. BCwb.These equations were used to calculate BCwb and BCff independently in an R script to double check the output from the instrument.
Line 215: What is SMAQMD, what ARB?We have defined these at first use, eg section 2.1 for SMAQMD. We have clarified the data sources in this paragraph as well.
Line 219 and Table 1: Is the R.M. Young systems instrumentation an ultrasonic wind instrument to measure the low wind speeds or is it a propeller Instrument? Should be mentioned in Table 1.Table 1 has been updated to reflect it is ultrasonic.
Line 236: Is PPD60PV-T2 a low cost sensor?This section has been removed, as the data are not discussed here and are presented elsewhere.
Line 239: Usually the manufacturers Default size Distribution has to be adapted to the local optical PM properties by comparison measurements?This section has been removed, as the data are not discussed here and are presented elsewhere.
Line 243: STL?This section has been removed, as the data are not discussed here and are presented elsewhere.
Line 250: FEM or FRM?This section has been removed, as the data are not discussed here and are presented elsewhere.
2.7.1 Statistical Methods: It would be better to describe the statistical Methods directly there where they are used (at the results). Then it would be better understandable when the application comes directly behind the description of the used statistical method. It avoids also double descriptions. This is also better for the lines 289 to 297.The style guide for the journal states that methods should be a separate section from the results; no action taken.
Line 302: There is a fsctor of 7.35 used to convert levoglucosan concentration to Wood Burning organic Carbon concentrations. Later Wood Burning PM is presented (e.g. in line 308 and in in Table 4). Is that calculated with the levoglucosan times the factor 7,35? Or how is Wood Burning PM calculated? See also lines 385/386.We have clarified this in the methods: the Puxbaum factor was used to determine wood burning PM concentrations as equal to 7.35 x levoglucosan.
Lines 309 to 315. This could be described within the results chapter when the Regression diagrams are shown.The style guide for the journal states that methods should be a separate section from the results; no action taken.
Results:
Headline: HAPS … full word!This has been updated to Hazardous air pollutants
Lines 348 to 359: In the field of Air Quality mesurements we talk About Uncertainties according ISO 20988:2007, not precision.The goal of this section is to quantify the precision of the measurements, not uncertainty.
Line 350/351: To calculate the Precision (better uncertainty) A and B are the concentrations from two collected samples. Are these sample collected at the same time and the same place with the same method? With other words Double Determinations?Yes, as described in the methods “collocated” is same place, time and method. We have reinforced that concept in this section by adding: “collocated (i.e., same place, time and method) samples”
Table 3: The concentrations are depicted here in ppb. They should be given here in µg/m3 according to the result's units in Table 5.We have updated this table to be in μg/m3.
Figure 2 is too small and the curves are partially not readable. Why not using the whole print space. DPM > write the full name like for the other two. Write the equations for the regression lines into the diagram.We have updated and enlarged this figure. Updates include spelling out “Del Paso Manor,” adding the full regression line equation in the diagram, and modifying colors the improve legibility.
I forgot what is BAM? Please, use a better understandable word for it.We have updated this sentence to simply say “hourly PM2.5”, and removed “BAM”
Line 414/415: The sentence is based on only 3 day time samples. Please, mention this as limitation for this message in the sentence.We have added the this caveat to the last sentence of the paragraph.
Line 419: Table 41? It is Table 4! The first line in the table is confusing: N 12-hr samples are all (total number) of samples. In brackets are the night time sample from these total number of samples. It would be even better to write the little number of daytime samples in brackets.Table number has been updated; N daytime samples have been added.
Figure 3: For better understanding the daytime samples should be named above the bars in the diagram. The hours on the x-axis are the starting times? This should be mentioned in the caption.Caption has been updated as requested. We have added an asterisk to indicate daytime samples.
Figure 4: The y-scale is not % of burning PM, it is burning % of PM2.5. In the lower diagram it is also PM2.5.Figure has been updated to PM2.5.
Lines 431 and 432 (Figure caption): ...during two daytime samples .... and two nighttimesamples according Figure 3.Caption has been updated.
Line 440: What is overall sample size?N has been added here.
Table 5: Where are the new site names described before? They should be marked and named in Figure 1. The pairs belonging together should be separated with bigger lines from each other.Figure 1 and this Table have been updated as requested.
Line 456: Daily patterns? where can we see these?We have revised this sentence to say “in the context of the diurnal BC, BCwb and BCff patterns”
Lines 457 ff: For the difference od daytime and nighttime is has also the stability of the atmosphere to be considered: Nighttime mostly stable, daytime instable (according the wind Speed in Figure 3).We have added text: day/night changes in the stability of the atmosphere will also play a role in concentrations”
41: Figure 5. For comparison it would be helpful if the bars of BCff and BCwb are directly side by side in one diagram.
We have updated this figure to show BCff and BCwb side by side.
Line 501: (see Supplemental Information Figures S-1-S-3). This would be good to show here, at least one of these diagrams.Done
Line 505: You can add: ...since People have mor time for heating.We have added: when residents tend to burn more
515 to 517: of cours dependent on the stability of the atmosphere, e.g. indicated by wind speed at these days according Fig. 3.We added a statement regarding day/night stability differences
Line 519: What is CBPF? It may be explained earlier but I forgot it and cannot find the explanation.CBPF is explained in section 2.7.1 in text and with an equation.
Line 522: The figure S-4 from the Supplemental Information would be helpful to be shown here.Given the length of figure S-4, it seems better placed in Supplemental, pending the editors’ opinion.
Line 533: The correlation of HAP with BCwb should be shown.We report regression results with R2 values in the text, and have added Supplemental Table 5 showing all R2 values for all HAPs, BCwb and levoglucosan
Lines 546, 547 and diagrams of Figure 6. These depicting of correlations are confusing. High correlation indicates little impact from the applied traces on the species in question, whereas correlation indicates high impact ....Here another way of correlation should be found where high correlation indicates a high impact and low correlation indicates a low impact according common sense.
We have clarified that the full results of the correlation analysis are now shown in Table 5, and added this table. We state that the wood burning tracers BCwb and levoglucosan have low correlation with the HAPs, indicating they do not have a common source. The plots in Figure 6 are ratio/ratio plots; the correlation in the plots on the left of Figure 6 indicate that the HAPs correlate no matter what the BCwb concentrations are. We have moved these to Supplemental, since the correlations in the new Table 5 are more straightforward to interpret.
Lines 578/579: Non- EJ communities have more device usage than EJ communities ... significantly higher number of burning days... In the seventh line of table 6 there it is just opposit?No, Table 6 shows that relative burn days for all categories are higher for non-EJ communities. We have simplified the statement in the paragraph prior to Table 6 to make this clearer.
Lines 9 to 13 in the table 6: What are Of burners?We have clarified this in the table as: “households that burn”
Conclusions, lines 593 ton 612: This text is more a summary than conclusions. In the conclusions should be explaine why have fossil fuel HAPs no significant differences, why is BCwb higher in Non-EJ communities etc. Such Points are concluded before in the textxs but not concluded here. This chapter should be a summary of conclusions not only a summary of results.We have clarified the conclusions that wood burning activity and concentrations are higher in non-EJ communities and that HAPs, BCff and mobile source emissions are higher in EJ communities.